# Biomarkers of Neurodegenerative Diseases: Biology, Taxonomy, Clinical Relevance, and Current Research Status

**DOI:** 10.3390/biomedicines10071760

**Published:** 2022-07-21

**Authors:** Dorota Koníčková, Kateřina Menšíková, Lucie Tučková, Eva Hényková, Miroslav Strnad, David Friedecký, David Stejskal, Radoslav Matěj, Petr Kaňovský

**Affiliations:** 1Department of Neurology, Faculty of Medicine and Dentistry, Palacky University and University Hospital Olomouc, 77900 Olomouc, Czech Republic; katerina.mensikova@fnol.cz (K.M.); lucie.tuckova@fnol.cz (L.T.); henykova.eva@centrum.cz (E.H.); miroslav.strnad@upol.cz (M.S.); david.stejskal@fno.cz (D.S.); petr.kanovsky@fnol.cz (P.K.); 2Department of Pathology, Faculty of Medicine and Dentistry, Palacky University and University Hospital Olomouc, 77900 Olomouc, Czech Republic; 3Laboratory of Growth Regulators, Faculty of Science and Institute of Experimental Botany of the Czech Academy of Sciences, Palacky University, 78371 Olomouc, Czech Republic; 4Laboratory of Inherited Metabolic Disorders, Faculty of Medicine and Dentistry, Palacky University and University Hospital Olomouc, 77900 Olomouc, Czech Republic; david.friedecky@upol.cz; 5Department of Laboratory Medicine, University Hospital Ostrava, 70852 Ostrava, Czech Republic; 6Department of Pathology and Molecular Medicine, Third Faculty of Medicine, Charles University and Thomayer University Hospital, 14059 Prague, Czech Republic; radoslav.matej@ftn.cz

**Keywords:** neurodegeneration, cerebrospinal fluid, biomarkers, blood-based biomarkers, neurodegenerative diseases, Alzheimer’s disease, Parkinson’s disease

## Abstract

The understanding of neurodegenerative diseases, traditionally considered to be well-defined entities with distinguishable clinical phenotypes, has undergone a major shift over the last 20 years. The diagnosis of neurodegenerative diseases primarily requires functional brain imaging techniques or invasive tests such as lumbar puncture to assess cerebrospinal fluid. A new biological approach and research efforts, especially in vivo, have focused on biomarkers indicating underlying proteinopathy in cerebrospinal fluid and blood serum. However, due to the complexity and heterogeneity of neurodegenerative processes within the central nervous system and the large number of overlapping clinical diagnoses, identifying individual proteinopathies is relatively difficult and often not entirely accurate. For this reason, there is an urgent need to develop laboratory methods for identifying specific biomarkers, understand the molecular basis of neurodegenerative disorders and classify the quantifiable and readily available tools that can accelerate efforts to translate the knowledge into disease-modifying therapies that can improve and simplify the areas of differential diagnosis, as well as monitor the disease course with the aim of estimating the prognosis or evaluating the effects of treatment. The aim of this review is to summarize the current knowledge about clinically relevant biomarkers in different neurodegenerative diseases.

## 1. Introduction

Neurodegenerative diseases are an extensive heterogeneous group affecting the nervous system. Pathophysiologically is characterized by selective involvement of the neural subpopulation of cells with a consequent clinical picture of the disease. A key element in the mechanism of neurodegenerative diseases is usually an overproduction of precursor protein associated with the formation of an aberrant protein with an unstable structure. This altered protein tends to form and accumulate intra- and extracellular protein clumps that may be related to neuronal death in the affected part of the brain. These clumps we defined as biomarkers [1].

The identification of biomarkers is currently a major topic in neurodegenerative diseases. Based on their definition, biomarkers are objectively measurable and evaluable indicators that serve to describe normal biological processes, pathological processes, and pharmacological responses to therapeutic intervention. The main goal in the use of biomarkers is to improve clinical diagnosis, in the sense of increasing the accuracy of differential diagnoses. Biomarkers should aid in estimating the stage of a disease and the rate of its progression, and they should be able to reflect therapeutic response rates. There are criterial differences between the sensitivity and specificity of biomarkers. Sensitivity reflects the success with which a test detects the presence of conditions being observed in a subject. Specificity reflects the ability to accurately select cases in which the examined features do not occur.

Biomarkers can be divided into groups according to the purpose they serve. The first group are diagnostic biomarkers. These might be subdivided into preclinical biomarkers, which can be used to detect a neurodegenerative process that is present but has not yet manifested with clinical signs; prodromal markers, which are useful when symptoms are present but not yet manifested in a way that allows the disease to be classified unambiguously; and clinical biomarkers, which are thought to help with the differential diagnosis of a disease and the symptomatology when very similar individual signs overlap. The second group are prognostic biomarkers, which serve in estimating the risk of disease development or in predicting the rate of disease progression. The third group are biomarkers that help to determine the phase or a stage of a disease. The fourth group are therapeutic biomarkers, used to monitor responses to therapeutic intervention [2].

The current concept of neurodegenerative diseases employs several categories of candidate biomarkers from the area of clinical signs; neurophysiology; biochemical indicators in cerebrospinal fluid (CSF), serum, or certain tissues; neuroimaging methods; and genetics. Methods of gene expression detection (microRNA) are increasing, in addition to classical gene analysis. Other scientific disciplines, such as proteomics and metabolomics, which deal with the study of functional molecules (proteins and neurotransmitter metabolites) potentially involved in neurodegenerative processes are also increasingly used [2,3]. The main effort is to create a characteristic profile, i.e., a combination of biomarkers that are specific to a given clinical unit or to a group of diseases sharing similar features.

CSF is considered an optimal source of biomarkers due to its direct relation to the extracellular space of brain tissue. It is assumed that all biochemical changes taking place in the brain are reflected in CSF [4]. The identification of CSF neurodegeneration biomarkers has been steadily improving for several years. Some biomarkers have been established and are being used in clinical practice, included in diagnostic criteria; other biomarkers are still being worked with to standardize analytic methods; and still other biomarkers are expected to have their meaning confirmed in future studies.

Neurodegenerative processes are connected with the abnormal aggregation and accumulation of pathologically altered proteins that are specific to given groups of diseases and are also reflected in the designation of proteinopathies. Simultaneously, they involve dysfunctions of a range of cellular mechanisms, including mitochondrial or lysosomal dysfunction, oxidative stress, and the response to inflammation caused by glial cell activation that ultimately results in neuronal degradation. These mechanisms are common for all disease units; their involvement in neurodegeneration is assumed to vary [3].

Similarly, CSF biomarkers can be distinguished. Specific biomarkers that reflect a type of accumulated pathological protein include amyloid beta (Aβ), phosphorylated tau protein (p-tau), alpha-synuclein (α-syn), transactive response (TAR) DNA-binding protein 43 (TDP-43), and prion protein (PrP^Sc^). Within non-specific CSF biomarkers, markers of axonal damage and axonal degeneration, including total tau protein (t-tau) and neurofilament light chains (NFL), have been the most studied. Among other non-specific markers, they are markers of synaptic, lysosomal, and mitochondrial dysfunction, proteins involved in the formation of intracellular proteins or participating in the degradation and clearance of abnormally modified proteins, and markers of glial activation (Table 1). Moreover, in some diseases, a ratio of two or more biomarkers is more specific and sensitive than a single biomarker.

This paper deals with the CSF biomarkers of the main groups of neurodegenerative diseases, some of which have already been introduced into clinical practice, others are still considered candidate biomarkers in ongoing research activities.

## 2. Alzheimer’s Disease

Alzheimer’s disease (AD) is the most common cause of dementia. Clinically, it is characterized by gradual cognitive deterioration with predominating episodic memory impairment.

The morphological basis is Aβ and tau protein accumulation in the form of amyloid plaques (senile plaques) and neurofibrillary tangles (NFT) in brain tissue (Figure 1). The current concept distinguishes several stages of AD: preclinical, prodromal, and clinical. During the preclinical stage, the presence of Aβ deposits in the brain is expected; however, none of the characteristic clinical signs are present yet. The prodromal stage is called mild cognitive impairment (MCI) with predominant episodic memory impairment when the cognitive disorder development does not yet meet the criteria for dementia. In the clinical stage, signs of cognitive dysfunction that interfere with the everyday activities of the affected individual are fully manifested [5]. This concept is particularly important in relation to treatment that might be able to modify the course of the disease—both to its development and testing in clinical stages and to its implementation and correct timing in clinical practice. In order to specify the clinical diagnosis of AD and its shift to the earliest possible stage of the disease, several biomarkers have been defined (hippocampal volumetry, FDG-PET, amyloid PET, and CSF biomarkers) as a part of the current NIA–AA (National Institute on Aging–Alzheimer’s Association) criteria for diagnosing and predicting late-onset AD and distinguishing MCI–AD from normal aging [6].

### 2.1. Amyloid Beta, Total Tau Protein, Phosphorylated Tau Protein

The primary biomarkers in AD are amyloid beta (Aβ), specifically its hyperphosphorylated isoform Aβ_42_, total tau protein (t-tau), and phosphorylated tau protein (p-tau), more specifically its phosphorylated form at position 181 (p-tau_181_). Aβ_42_ reflects cortical amyloid deposition, t-tau reflects neurodegenerative process intensity, and p-tau correlates with the intensity of neurofibrillary changes. AD is characterized by the combination of a significant decrease in CSF Aβ_42_ and significant increases in CSF t-tau and p-tau. This combination serves as a clinical biomarker to determine a diagnosis as early as at the stage of full-blown dementia and as a prodromal biomarker at the stage of MCI, when with their help, it is possible to predict the conversion of MCI to AD. Determining t-tau/Aβ_42_ or p-tau/Aβ_42_ correlation ratios is also recommended [4].

Within differential diagnosis, these biomarkers can be useful for differentiating AD from other diseases manifested with progressive cognitive dysfunction, such as progressive supranuclear palsy (PSP) and corticobasal degeneration (CBD), but also from depression, which may mimic a cognitive deficit. With the given clinical units, CSF levels of these proteins are normal. The P-tau level has been suggested as a possible diagnostic biomarker for differentiating AD from frontotemporal dementia (FTD) and dementia with Lewy bodies (DLB). In the context of differential diagnosis, usage of CSF Aβ_40_ is suggested as well. Aβ_40_ is the prevailing isoform of Aβ, with which no significant changes in CSF levels between AD and healthy controls have been identified. Increased CSF concentrations of Aβ_40_ were observed with AD as compared to Parkinson’s disease dementia (PDD) and DLB, implying that the Aβ_42_/Aβ_40_ ratio might better help to distinguish than Aβ_42_ alone between AD and non-AD dementias: mainly PDD, DLB, and vascular dementia [7].

### 2.2. Neurofilament Light Chains (NFL)

Neurofilaments are the main structural components of long myelinated axons. Their high concentrations in CSF generally reflect rapidly progressive neurodegenerative processes. In contrast to FTD and atypical Parkinsonian syndromes, their CSF concentrations are normal in AD. As such, they can be used in differential diagnoses to distinguish AD from other diseases accompanied by dementia [5,8].

### 2.3. Other Candidate CSF Biomarkers of AD

In addition to the key CSF biomarker examination methods that have already been standardized, many other indicators are being researched that could be able to objectively measure other pathological processes involved in AD.

β-site APP-cleaving enzyme 1 (**BACE1**) is a β-secretase involved in the formation of Aβ from amyloid precursor protein (**APP**), and both its activity and its concentration in CSF increase in AD as early as the MCI stage [7].

**Neurogranin** and synaptosomal associated protein (SNAP25) have been labelled candidate biomarkers of synaptic dysfunction; their high concentrations in CSF may predict progression to AD in patients with MCI correlating with the rate of cognitive deterioration. Their CSF concentrations together with another biomarker, visinin-like protein 1 (VILIP-1) were significantly higher with AD than with healthy controls or with other types of neurodegenerative dementias [7,9].

**Aβ oligomers** are considered another promising biomarker; they are regarded as its toxic form, inducing a synaptic response, and their higher concentration in CSF is associated with AD. Currently their examination is rather limited, due to very low concentrations in CSF and thus only a complicated quantification with the help of an enzyme-linked immunosorbent assay (ELISA). The introduction of new ultrasensitive methods into laboratory diagnostics could be helpful [10].

## 3. Synucleinopathies (LBD, MSA)

Synucleinopathies are a group of diseases characterized by the accumulation of pathologically modified protein alpha-synuclein (α-syn), in various parts of the central and peripheral nervous system. Alpha-synucleinopathies include (1) Lewy body diseases (LBD) including Parkinson’s disease (PD), Parkinson’s disease dementia (PDD), dementia with Lewy bodies (DLB), and (2) multiple system atrophy (MSA). Individual clinical entities differ from each other due to their clinical symptomatology, the distribution of pathological changes, and the type of cells affected.

LBD are pathologically characterized by neuronal intracytoplasmic inclusions known as Lewy bodies and Lewy neurites (Figure 2); MSA is primarily associated with oligodendroglial cytoplasmic inclusions (Figure 3).

However, the most important part of all the mentioned inclusions is pathologically modified α-syn [11]. α-syn is a protein composed of 140 amino acids, expressed mainly in neurons, specifically in neuronal synapses. In its monomeric state, it is soluble and represents the most common form found in cytoplasm. Under pathological circumstances, its aggregation into oligomers occurs, later into fibrillar aggregates. Its oligomeric forms are considered to be the most neurotoxic [12]. Clinical entities from the synucleinopathies share a range of symptoms not only with each other, but also with a number of other neurodegenerative proteinopathies. For this reason, differential diagnosis, mainly in the early stages of disease, is very complicated as no clear and reliable biomarker has yet been identified. The strongest association to all synucleinopathies is probably that of REM sleep behavior disorder.

PD is also described with hyposomnia, depression, and obstipation; DLB is accompanied by early recurrent hallucinations; MCI occurs with predominant visual-spatial orientation disorder. Generally available imaging methods (3T-MRI, DaTscan) lack sufficient sensitivity and specificity to distinguish between PD and other Parkinsonian syndromes, and transcranial sonography cannot reliably make this distinction either [3]. Examinations using more sophisticated methods could help, but they are very expensive, and their use is limited to several research centers. For this reason, analyses to identify biochemical markers in CSF, serum, skin, nerves, intestinal mucosa, or saliva are offered [12].

Among CSF biomarkers, concentrations of α-syn and its various forms, CSF level Aβ, tau-protein, and neurofilament have been studied in greater detail due to their possible use in clinical practice.

### 3.1. Alpha-Synuclein and Its Forms

In synucleinopathies, a decrease in CSF α-syn is generally observed. This is mostly explained as a result of its increased aggregation and accumulation in the central nervous system. The level of total α-syn (t-α-syn) can be used to differentiate synucleinopathies from other proteinopathies; however, it is not useful for differentiating between individual synucleinopathies, even though significantly lower concentrations have been observed in MSA than in PD and DLB [13].

Other forms of α-syn can also have differential diagnostic value within synucleinopathies: oligomeric (o-α-syn), phosphorylated (p-α-syn), and pro-aggregated α-syn. The levels of o-α-syn are higher in PD and DLB. Higher concentrations of p-α-syn were observed with PD than with controls, MSA, or PSP [3,14].

The detection of pro-aggregating α-syn forms, with the help of the ultrasensitive real-time quaking-induced conversion method (RT-QuIC), is standardly used to diagnose prion diseases and has the potential to diagnose a disease in its presymptomatic stage [15]. Using this method, it was possible to detect α-syn aggregates in the central nervous system in REM sleep behavior disorder patients who developed synucleinopathy at a later stage. RT-QuIC also helped to determine α-syn aggregates in CSF in patients with DLB with 92% sensitivity, with PD with 95% sensitivity, and with a total of 100% specificity compared to AD and healthy controls [16]. However, more data are needed to verify the sensitivity and specificity of this ultrasensitive method.

### 3.2. Beta-Amyloid, Total Tau Protein, Phosphorylated Tau Protein

CSF concentrations of these biomarkers used in AD diagnostics are within the normal range in PD [17]. Decreased Aβ_42_ levels in PD are considered a predictor of the development of cognitive deficit and subsequent PDD. From the perspective of tau protein, CSF levels of t-tau and p-tau in PD were lower than in DLB, PDD, AD, and MSA in a number of studies. Higher levels of tau protein might be, similarly to decreased Aβ_42_, a predictor of the development of cognitive deficit [3,18].

More attention has been paid to these biomarkers in DLB. DLB patients show decreases in the CSF levels of Aβ_42_ that are comparable with AD. However, the decrease in Aβ_42_ in DLB occurs only during the development of synucleinopathy when amyloid deposits are produced; in AD, it is possible to detect its very low levels already in the prodromal stage of the disease. Unlike in AD, decreased Aβ_40_ levels were observed in DLB. For this reason, the Aβ_42_/Aβ_40_ ratio can be suitable for distinguishing between these two entities. T-tau and p-tau levels are significantly lower in DLB patients than in AD patients but higher than in healthy controls or PD [5].

### 3.3. Neurofilament Light Chains (NFL)

Neurofilaments are the main structural component of an axon, generally considered a biomarker of degeneration of long myelinated axons; their high concentrations in CSF are regarded as a correlate of a rapidly progressive neurodegenerative process.

Their CSF levels do not increase in PD, mainly because of a less severe axonal degeneration. As a probable reflection of the degeneration of long myelinated axons and the increase in CSF NFL present in PSP, MSA, CBD, they can be useful CSF markers contributing to differentiating PD from these forms of atypical parkinsonism [19].

### 3.4. Other Potential Markers in Synucleinopathies

In the group of synucleinopathies, attention is paid to biomarkers that mainly play a role in mitochondrial dysfunction, oxidative stress, and lysosomal dysfunction.

Protein DJ-1 is a multifunctional protein involved in a number of cellular processes; its decreased function leads to oxidative stress. At first, results of studies dealing with CSF concentrations of DJ-1 in neurodegenerative diseases were controversial. Using a new, highly ultrasensitive *Luminex assay*, decreased CSF levels of DJ-1 were found in PD compared to controls, AD, and MSA [20].

Ubiquitin C-Terminal Hydrolase-L1 (UCH-L1) is a protein participating in the degradation of abnormally modified proteins from neuronal cytoplasm. A significant decrease in its CSF levels was found in PD compared to PSP or MSA [3,21].

β-glucocerebrosidase (GBA) is a lysosomal hydrolase involved in α-syn degradation. Its decreased amount due to a mutation in the *GBA1* gene is considered the major risk factor for PD. Decreased CSF levels of GBA have been mainly found in the early stages of PD. As such, CSF activity together with o-α-syn/t-α-syn ratio has been suggested as a candidate diagnostic biomarker of early PD [13].

The levels of the dopamine metabolites dihydroxyphenyl acetate (DOPAC) and homovaleric acid (HVA) are being further examined, whereas continuous research aimed at different microRNA (miRNA) may also bring promising results. miRNAs are single-stranded chains of non-coding RNA involved in regulating the gene expression of almost all target genes associated with PD. They are assumed to be tissue-specific, abundant, highly stable, and quantifiable. Their upregulation or downregulation occurs several years before the onset of PD; thus, they could serve as biomarkers of the early stages of PD [2,3].

## 4. Tauopathies, TDP-43 and Other Rare Proteinopathies (Frontotemporal Lobar Degeneration)

Frontotemporal lobar degenerations (FTLD) are a clinically and neuropathologically heterogenous group of diseases. Their common denominator is dominantly affected frontal and temporal lobes; the basal ganglia and parietal cortex are also often affected. Due to the neuropathological background, their current classification separates into two major groups: tauopathies (FTLD-tau) and non-tau FTLD [22].

FTLD-tau are characterized by intraneuronal and/or glial deposits of tau protein. They are generally referred to as “primary tauopathies” (Figure 4). In relation to the predominant isoform of tau protein, which is determined by its posttranslational modifications, they are further subdivided into subtypes 3R and 4R.

Among non-tau FTLD, the most common present deposits are those of TDP-43 protein (Figure 5); deposits of ubiquitin or FUS protein are rarer. Based on this, groups are then further referred to as TDP-43 proteinopathies (FTLD-TDP), ubiquinopathies (FTLD-U), and FUS proteinopathies (FTLD-FUS) [23].

The clinical manifestation of FTLD can generally be divided into three main syndromes, yet these frequently overlap and vary in their dominance, which are determined by the density and distribution of the given tau or non-tau pathology. FTLD can manifest as the behavioral variant of frontotemporal dementia syndrome (*bv*FTD), primary progressive aphasia syndrome (PPA), or a combination of dementia with movement disorders which can be characterized by extrapyramidal symptoms or motor neuron diseases [22].

The primary tauopathies (FTLD-tau) currently include entities related to extrapyramidal symptoms, including FTD and parkinsonism linked to chromosome 17 (FTDP-17), progressive supranuclear palsy (PSP), and corticobasal degeneration (CBD). Other diseases to rank among them are Pick’s disease, nonfluent variant primary progressive aphasia (*nfv*PPA), chronic traumatic encephalopathy (CTE), and several entities that have only been pathologically defined, the phenotypes of which have not yet been described unambiguously, including argyrophilic grain disease (AGD), aging-related tau astrogliopathy (ARTAG), globular glial tauopathy (GGT), and primary age-related tauopathy (PART) [22].

TDP-43 proteinopathies typically manifest as a combination of dementia and motor neuron disease for which there is a conventional term: frontotemporal spectrum disorder in amyotrophic lateral sclerosis (ALS). They can also manifest with *bv*FTD syndrome or the semantic variant PPA (*sv*PPA) [23].

Motoneuron diseases (MNDs) are a group of progressive neurodegenerative disorders characterized by the loss of motor neurons in the cortex, brainstem and spinal cord. They generally cause weakness without significant sensory symptoms or pain. The most common form is ALS, which affects both upper and lower motoneurons [24]. The diagnosis of ALS is mainly based on clinical examination and electromyography. Sometimes the diagnosis of the disease is delayed as the initial symptoms may mimic other diseases [25]. Validated biomarkers that can facilitate earlier diagnosis of ALS or provide early treatment of the disease, are urgently needed. CSF biomarkers may point to neuronal damage in the early stage of the disease. Neurofilament light chains, phosphorylated neurofilament heavy chains (pNFH), chromatogranin A, and TDP-43 protein are the most prominently studied as markers [24,25].

A number of the mentioned clinical entities have a defined clinical picture that typically develops only at later stages of the disease. A similar situation is found with imaging methods, when characteristic findings occur in images but are not necessarily unambiguously expressed in the early stages of a disease; however, more sophisticated imaging methods are not used in routine practice. For this reason, close attention in this group of diseases is paid to identifying CSF biomarkers that could serve in differential diagnoses in the early stages of the disease and possibly also as prognostic or therapeutic biomarkers. From the perspective of differential diagnostics, in current clinical practice and with respect to prognosis and therapeutic approaches, it is mainly important to distinguish *bv*FTD from the frontal variant of AD (*fv*AD), and to distinguish psychiatric disorders mimicking *bv*FTD [22], PSP, and CBD from other forms of dementia and Parkinsonian syndromes (PD, LBD, MSA).

### 4.1. Beta-Amyloid, Total Tau Protein, Phosphorylated Tau Protein

CSF concentrations of these AD-specific proteins do not change radically in FTLD. However, they can be used to distinguish between *bv*FTD and *fv*AD, or to distinguish the logopedic variant of PPA from two other forms of PPA. In *fv*AD and *lv*PPA, there are decreased levels of Aβ_42_ and increased t-tau and p-tau levels [22]. However, these biomarkers are not able to distinguish FTLD from psychiatric disorders of non-degenerative etiology. In FTLD-tau, no specific markers of tau pathology have yet been identified; CSF t-tau and p-tau are generally lower in FTLD-tau compared to AD and FTLD-TDP, yet very variable in individual entities [26,27]. However, studies of tau protein fragments have not shown any specific biomarker for this FTLD group. The RT-QuIC method could be a major asset; this method is currently being tested both in 3R and 4R tauopathies [28,29,30]. However, due to the biochemical complexity of tau protein and its isoforms, this method is not yet fully ready for routine clinical practice.

### 4.2. TDP-43

FTLD-TDP (TDP-43 proteinopathy) is a major subgroup of non-tau FTLD. TDP-43 immunoreactivity, together with the presence of this pathology in brain tissue, is reflected in higher concentrations of TDP-43 in CSF. CSF levels of TDP-43 were higher than in heathy controls, but compared to FTLD-tau the values overlapped [26]. Comparing CSF TDP-43 patients with pathology-confirmed diagnoses of AD, FTLD-tau, FTLD-TDP, and prion diseases, significantly increased TDP-43 levels were found in the prion diseases group. However, when comparing the remaining groups, no noticeable differences were proven [27]. Therefore, the diagnostic accuracy of TDP-43 is limited, mainly due to a lack of highly sensitive quantitative methods. The evaluation of TDP-43 protein in blood exosomes from the brain, which likely reflect brain pathology better, could be beneficial in the future. Its predictive relevance could also increase in combination with other biomarkers [26].

### 4.3. Neurofilament Light Chains (NFL)

NFL levels in FTLD are significantly increased compared to AD. Within individual forms of FTLD, NFL concentrations show large variability; therefore, they are not a reliable marker to differentiate individual clinical entities. However, they can be of importance as a marker to distinguish *bv*FTD from primary psychiatric disorders that mimic them, as well as a prognostic indicator to predict progression to the clinical stage in presymptomatic mutation carriers (concentrations rapidly rise immediately before the onset of clinical signs), or to predict a progression rate in the course of a symptomatic stage of a disease [26].

### 4.4. Progranulin

Progranulin is a precursor of granulin, the protein encoded by the *GRN* gene. Mutations in *GRN* genes are among the most frequent mutations associated with TDP-43 proteinopathies. Most pathogenic variants in this gene result in increasing progranulin levels, measurable in both serum and in CSF. Measuring progranulin levels in serum can thus be used as a less expensive way to detect pathogenic mutation than genetic testing.

Progranulin levels in serum and in CSF depend on the type of *GRN* gene mutation.

According to the rate of reduction, we can therefore estimate probable pathogenicity, i.e., whether it is an unambiguous pathogenic variant or mutation that represents an increased risk of disease development. Pathogenic mutation carriers have significantly lower progranulin levels than FTLD, whereas they have a much more severe course of neurodegeneration. In addition, decreased progranulin levels are detectable with 100% sensitivity and specificity even in presymptomatic individuals [26].

### 4.5. Dipeptide Repeat Proteins (DPRs)

Recently described mutations in the *C9orf72* gene are the next most frequently reported mutations associated with *bv*FTD, mainly in the form associated with symptoms of motor neuron diseases. DPRs are products of *C9orf72* gene mutation, considered the key toxicity mechanism of this mutation. Due to a mutation that means GGGGCC hexanucleotide repeat expansion, five DPRs are produced; the levels of only one of them, poly (GP, are measurable, whereas its increased concentration was found in symptomatic and presymptomatic individuals [31]. Concentrations of this protein are not currently determined, but it could serve the same function as progranulin in detecting expansion before genetic testing. This protein could be a useful biomarker in measuring therapeutic response [26].

### 4.6. FUS Protein

FTLD-FUS are a rare group of FTLD. No study or method have yet proved the presence of FUS protein in CSF [26].

## 5. Prion Diseases

Prion diseases are rapidly progressing diseases characterized by an accumulation of misfolded prion protein PrP^Sc^ in brain tissue (Figure 6).

The most common form is sporadic Creutzfeldt–Jakob disease (sCJD). It manifests with heterogenous symptoms with a dominating rapidly progressive cognitive deficit. Within clinical diagnosis, several biomarkers have been established that are able to increase its accuracy. Some have been gradually included into clinical diagnostic criteria; some serve as supportive ones. These are combinations of specific EEG and brain MR findings, together with testing certain parameters in CSF and the *PRNP* gene sequence [32]. PRNP mutations account for 10–15% of all human prion diseases. Some cause specific clinical syndromes (Gerstmann–Sträussler–Scheinker syndrome (GSS) and fatal familial insomnia (FFI)), whereas others may mimic the clinical manifestation of CJD. *PRNP* sequencing is therefore an important biomarker that should be considered in the differential diagnosis of prion diseases and is crucial in cases of atypical, rapidly progressive dementias [15].

Until recently, protein 14-3-3, t-tau or its alternative, and the p-tau/t-tau ratio have been considered the main biomarkers. Other identified biomarkers with lower diagnostic accuracy include S100b protein, total prion protein (t-PrP) and α-syn. As in other rapidly progressive neurodegenerations, neurofilaments also play a role. A major breakthrough in *ante-mortem* diagnostics was presented in 2011, with the development and introduction of the PrP^Sc^ amplification technique RT-QuIC into clinical practice. CSF RT-QuIC and/or RT-QuIC from a nasal swab have been included in the current diagnostic criteria of sCJD, thanks to its reliability and diagnostic accuracy in sCJD, as proven in a number of studies with a specificity of 99–100% [33,34]. It exhibits high sensitivity for genetically caused prion diseases, namely in patients carrying the E200K and C210I mutation, but has a low detection sensitivity in discrimination of GSS and FFI. As the most significant exception is that patients with GSS carrying the p.D202N substitution consistently showed early extrapyramidal features, it often raised the suspicion of atypical parkinsonian syndrome [15,35].

### 5.1. RT-QuIC Analysis

CSF RT-QuIC analysis is a method able to directly detect pathogenic protein; it is very sensitive, even in the early stages of the disease. If available, its usage is recommended in all cases with a suspected prion disease [15]; however, technical complexity and financial concerns complicate the wider use of this ultrasensitive method.

### 5.2. 14-3-3 Protein and Total Tau Protein

14-3-3 protein and total tau protein are very sensitive biomarkers, but they are not specific for prion diseases. 14-3-3 protein can be increased within other afflictions, such as brain injury or encephalitis. Tau protein, contrastingly, might not reliably distinguish rapidly progressive cognitive deficit in atypical AD, since the level of total tau in CJD are much higher. For this reason, these biomarkers should not be used in the context of general screening, but in a relevant clinical context when a prion disease is suspected [15,27].

### 5.3. Neurofilament Light Chains

Several studies proved the high diagnostic accuracy of NFL in distinguishing sCJD from healthy controls, yet their specificity in distinguishing sCJD from other rapidly progressive neurodegenerative dementias was insufficient for routine usage [15].

## 6. Biomarkers of Neurodegenerative Diseases

Neurodegenerative diseases are characterized by the interaction of numerous molecular pathways that can be assessed through bodily fluids, particularly cerebrospinal fluid (CSF) and blood. The ideal biomarker for early diagnosis should be sensitive and specific for early neurological changes, able to discriminate between the disease state and conditions of normal aging. Based on this description, peripheral markers play an important role in solving diagnostic limitations, provide alternative and non-invasive solutions for identifying a disease, and are considered indicators of cognitive and biological changes in the brain [36].

CSF is one of the most useful biological fluids, directly reflecting biochemical processes in brain tissue. It represents an ideal matrix for providing a biological fingerprint of the central nervous system (CNS), and it enables an early pre-clinical diagnosis of a disease. However, monitoring neurodegenerative diseases requires a less invasive and more accessible option, such as blood plasma or serum [26,37].

Blood communicates with the brain via the hematoencephalic barrier, lymphatic vessels, and glymphatic system. However, several problems arise when analyzing blood:(1)A biomarker originating in the CNS has to cross the hematoencephalic barrier to be detectable in the periphery; if its concentration in CSF is low, it will be even lower in the blood as a result of the volume ratio of CSF, causing substantial dilution of the analyte;(2)If a biomarker is not specific to the CNS but is expressed in the peripheral tissues, the CNS contribution is potentially lost against the high biological background caused outside the CNS;(3)The presence of vast amounts of other proteins in the blood (albumins, globulins, transferrin, antitrypsin, fibrinogen, etc.) can result in analytical problems due to their possible interference;(4)The presence of heterophilic antibodies in the blood may interfere with immune tests;(5)An observed analyte can be a subject of proteolytic degradation in the plasma;(6)Pre-analytical changes may exist [38].

## 7. Biomarker Analysis

Measuring biomarkers of neurodegenerative diseases in the blood presents a number of challenges that require sensitive and specific tests [39]. As a result of the selectivity of the hematoencephalic barrier, which prevents the free passage of molecules between the CNS and blood compartments and affects the volume ratio of blood to CSF, concentrations of CNS-derived proteins are much lower in blood. Non-invasive and cost-effective examinations such as a biomarker analysis can be used for the initial screening phase within a multistage diagnostic process to differentiate neurodegenerative diseases [26].

The heterogeneity of neurodegenerative diseases impedes the development of early diagnosis and effective medical procedures as well as therapeutic interventions. Omics techniques contribute to a more complex view of molecular pathways underlying the development of neurodegenerative diseases and help to discriminate between individual subtypes on the basis of specific molecular features. These techniques include detecting disease-related DNA sequence variants (genomics), transcriptome and non-coding RNA profiling (transcriptomics), genome-wide identification of DNA-protein interactions (epigenomics), analyses of protein–protein interactions (proteomics), and analyses of metabolic systems (metabolomics).

Most human diseases, including neurodegenerative diseases, are multifactorial diseases characterized by a number of molecular aberrations that act in a concurrent or synergistic manner in the course of a disease. Omics analyses are only able to detect components of a particular pathway, and have limited prognostic or therapeutic value. At the same time, they are able to identify molecular aberrations at various levels (gene, transcript, protein synthesis, and post-translational modifications, cellular metabolic processes, etc.), maximize available information, and increase the possibility of identifying the principal causes of the disease.

Multi-omics analyses and characterizations of patients’ omics profiles allow deeper examinations of neurodegenerative diseases, providing a more comprehensive perspective of multifactorial disorders that supports the development of specific targeted and personalized therapies [40]. If they meet conditions of reproducibility and easy repeatability, they identify the ideal biomarkers for measuring disease progression and for monitoring responses to disease-modifying therapies [41].

A potential method of peripheral identification of neuron-derived biomarkers is to measure proteins in exosomes. This technique makes it possible to measure the content of nucleic acids and proteins in extracellular vesicles, offering a less invasive approach to CNS proteome [26]. However, the sensitivity of the ELISA method is not sufficient for identifying and reliably quantifying the concentration of CNS biomarkers in plasma or serum. Fortunately, advanced and highly sensitive techniques have markedly improved the detection of peripheral biomarkers in technologies related to the enhanced availability of ultrasensitive and more sensitive analytical methods, such as the single molecule array (SIMOA) method and RNA sequencing (RNA-Seq) microarrays, allowing the bridging and eliminating of many obstacles so that blood biomarkers have a promising future [42]. They represent a valid tool for defining target engagement as well as for long-term monitoring of treatment results [38].

## 8. Exosomes

All types of cells in the CNS are able to release extracellular vesicles (EV), including neurons, astrocytes, oligodendrocytes, and microglia. EVs are small vesicles released by all cells. We distinguish two types: the first type, widely known as exosomes, is produced and accumulated in a vacuole. The second type is ectosomes, which are released by direct budding from the plasma membrane. Both types share common properties. EVs of neuronal and astrocytic origin isolated from blood can thus be used to examine brain pathological processes.

Exosomes are vesicles of nanomolar size, released by eukaryotic cells derived from endosomes. They are considered vesicles through which donor cells transfer proteins, lipids, and nucleic acids into target cells, thereby influencing metabolism. It has been proven that they participate in physiological processes in vivo, including intercellular communication, cell migration, angiogenesis, and anti-cancer immunity [43]. Moreover, they can be involved in inflammatory processes, which play a key role in a huge number of pathological conditions, including neurodegenerative diseases. The association between inflammation and changes in the nature or level of certain exosomal contents is a fundamental step toward identifying new potential biomarkers. Thanks to their unique structure and function, they have several different and potentially important applications. The advantage of using exosomes as marker vesicles lies in their simple collection from a bodily fluid such as blood, urine, or saliva; they represent non-invasive options for pathology screening. However, they only became a potential for research based on discovery of their content—functional proteins, messenger RNA (mRNA), and micro RNA (miRNA).

Increasing evidence indicates that exosomal miRNAs play important roles in neurodegenerative disease progression. They contribute to neurodegenerative processes primarily in three ways: (1) targeting mRNA genes related to regulating or inhibiting protein translation or degradation; (2) participating directly in neuroinflammation via toll-like receptor binding or via regulation of mRNA expression; and (3) by causing a defect in miRNA production itself. Neurons can transport miRNA via exosomes into astrocytes and thus indirectly regulate protein expression in them [44]. Proteomics studies after exosome isolation through immunoadsorption have revealed that exosomes originally contain proteins localized in endosomes, plasma membrane, and cytosol; only a few components are from the nucleus, mitochondria, endoplasmic reticulum, or Golgi apparatus. This implies that they can mediate intercellular communication and reflect cellular changes that provide responses to pathological conditions. Based on these findings, they are therefore considered potentially effective sources of biomarkers for observing disease progression [45,46].

In vitro studies show a huge contribution in the modulation of synaptic activity, as they regulate a great number of postsynaptic receptors, neurotransmitter release, and neuronal excitability. The functioning of EV biomarkers, based on multiple characteristics of proteins and miRNA, has offered sufficient evidence suggesting their enormous potential as a diagnostic, evaluative, and therapeutic tool [47]. In addition to the usefulness of exosomes and EVs as biomarkers, these molecules can be used as target molecules or therapeutic drug carriers. Their significance has several qualities: (1) they are highly stable in the circulatory system and can therefore have a long-distance range; (2) they can cross the hematoencephalic barrier; (3) they transport molecules into the CNS, suitably mainly for treatment of neurodegenerative diseases; (4) they have a high affinity to tissues; (5) they decrease the risk of non-target effects and avoid phagocytosis or degradation; (6) they have high biocompatibility and poor immunogenicity and toxicity; and (7) they show high availability as they are easily obtainable from bodily fluids [48].

## 9. Parkinson’s Disease and Dementia with Lewy Bodies (PD and DLB)

Parkinson’s disease is characterized by histopathological depositions of Lewy bodies (LB) and Lewy neurites, which contain, among other proteins, misfolded α-synuclein. Although LB are first found at the periphery (e.g., in the olfactory bulbus), they gradually spread into the brain stem, limbic system, and finally into the cerebral cortex. The diagnosis of PD is still based on clinical observation; the diagnosis is rather limited due to the wide spectra of symptom, most of which are similar to those of some other neurodegenerative disease [43].

### 9.1. The Role of α-Synuclein

The role of α-synuclein in the peripheral tissue and blood is still being discussed. Initially, α-synuclein was considered to be an intracellular protein, yet its identification in CSF and plasma prove that it is secreted by cells. It aggregates to form monomeric and oligomeric forms, protofibrils and fibrils; its concentration is thus significantly decreased [39]. Most human α-synuclein is highly expressed in erythrocytes and even slight blood contamination as a result of traumatic lumbar puncture can significantly change its quantification in CSF. The increased concentration of oligomeric form in blood therefore emerged as one option of a peripheral biomarker of PD. As recently described, the contamination of blood platelets and hemolysis can also significantly affect levels of non-specific diagnostic marker proteins, such as DJ-1 (daisuke-junko-1) in serum or plasma. In relation to this, a large number of biological, epidemiological, and clinical studies have described that the incidence of urate in serum decreases the risk of PD. Since urate is considered to be an important endogenous antioxidant, its increased levels can contribute to fighting oxidative stress and PD pathogenesis. Its high levels can also imply a lower risk and slower progress of the disease, making it a promising biomarker for the diagnosis and prognosis of PD [37,49,50].

### 9.2. miRNA as Potential PD Biomarkers

miRNA molecules are considered a promising blood biomarker for neurodegenerative diseases. These are 22 nt small non-coding RNAs which bind to the 3′-untranslated region of target mRNAs while they direct post-transcriptional repression of target genes by producing an RNA-induced silencing complex, which leads to the destabilization or translational inhibition of mRNA. Some miRNAs are encapsulated microvesicles, present in a relatively stable form in bodily fluids, including serum or plasma [38]. miRNA-124 is one of the most frequently expressed miRNAs in the brain involved in neurogenesis, the morphology of synapses, neurotransmission, inflammation, autophagy, and mitochondrial function. In addition, clinical evidence indicates that decreased levels of miRNA-124 in plasma can serve as a potential diagnostic biomarker in PD. In most studies, the downregulation of miRNA was observed in the early stages of neurodegeneration, implying that its reduction may not only reflect dopamine-induced cell death, but may even contribute to the initial biological process of neurodegeneration in PD. As nigrostriatal degeneration also substantially influences nicotinic receptor-mediated dopamine signaling and miRNA-124-mediated inhibition of nuclear factor kappa light chain enhancer of activated B cells (NFkβ), this could be a suitable strategy for enhancing protective anti-inflammatory cholinergic effects [51].

### 9.3. Difference between PD and APD

It is often difficult to distinguish between PD and atypical parkinsonian disorders (APD), particularly during the early stage of the disease. APD are a heterogenous group of diseases, part of which include pathologically-changed alpha synuclein, as does PD. The most important synucleinopathy in the APD group is multiple system atrophy (MSA). Currently, no blood test in clinical practice is able to distinguish between PD and APD.

α-synuclein is present in peripheral fluids; however, its concentration is strongly affected by the presence of red blood cells. Therefore, the use of total free α-synuclein is limited by erythrocyte contamination. An alternative approach involves quantifying the α-synuclein released from neural tissue. The results of this study indicate the measurement of neuronal exosomes in serum connected to α-synuclein and clusterin function as the best predictive PD marker, in contrast to atypical parkinsonism, including MSA and prodromal PD. Measuring its content can thus be used as a substitute for its intraneuronal processing, and therefore as a potential marker for monitoring modifications of disease therapy, especially in its early stages [52].

### 9.4. NFL as Diagnostic PD Biomarkers

Neurofilament light chains (NFL) are one of the three subunits of a neurofilament group. They are specific cytoskeletal neuronal proteins, well-represented, particularly in unmyelinated axons. Damage to axons causes the release of NFL into CSF and blood [53]. As blood NFLs have a diagnostic accuracy similar to CSF NFLs, they may be a valid clinical tool in differential diagnosis between PD and APD [54]. Increased NFL levels were observed in patients with PD in connection with severe and long-term disease, in the context of the presence of dementia [55]. Although NFL in CSF and blood can differentiate between PD and APD, they cannot be used as a tool to differentiate between APD with synuclein depositions (MSA), and APD related to the depositions of pathologically changed protein tau (tauopathies) as progressive supranuclear palsy (PSP) and corticobasal degeneration (CBD). Study outcomes have indicated that NFLs correlate with severity and the motor and cognitive progression rate of a disease, yet not with its duration. This finding is supported by the claim that NFLs could represent a powerful predictor of progression across parkinsonian phenotypes [55,56].

### 9.5. PD and Exosomes

Exosomes have two main roles in PD pathogenesis: (1) they are the principal mediators of α-synuclein cell-to-cell transmission, and (2) they have the ability to transport RNA, mainly miRNA [57]. Several experiments confirmed that α-synuclein is secreted directly into the extracellular space, or it is transmitted via exosome pathways while its secretion processes are regulated with intracellular calcium concentration.

In PD, exosomes containing α-synuclein are released by damaged neurons. They can change its spread from original neuron-to-neuron to spread neuron-to-glial cell spread; this leads to an inflammatory response through the activization of microglial cells [46]. Although α-synuclein levels in exosomes are low, recent studies showed that exosomes provide an ideal environment for its aggregation, and thus potentially promote its spread and PD pathology. This has also been supported by the finding that oligomeric forms of α-synuclein are more easily accepted by recipient’s cells than their free forms [44]. More detailed research has proven that exosomes contribute to non-cell autonomous mediation of neurotoxicity, which allows the long-range transportation of α-synuclein, as well as other mechanisms of intercellular communication and intracellular changes that occur in reaction to pathological conditions, and are an effective measure of disease progression [45,57].

### 9.6. Influence of Proteomics on PD

The study of proteomics employs extensive techniques aimed at detecting, identifying, and characterizing protein or peptide changes. Proteomics can distinguish between different protein isoforms and detect post-translational modifications that may be of functional significance, and thus can serve as a matrix for understanding the underlying molecular mechanisms and biomarker development for early detection. The proteomic approach is now emerging as a promising platform for biomarker discovery, due to its accuracy, specificity, and sensitivity in diagnosing and monitoring disease progression, which remains a challenge in PD.

A review by Chelliah et al. [58] highlighted the blood biomarkers of PD that have been systematically analyzed based on their distinct ability to differentiate between PD patients and healthy controls. The authors included apolipoprotein A-I (ApoA-I), haptoglobin, clusterin, Inter-α-trypsin inhibitor heavy 4 (ITIH4), and transthyretin in this study. Apolipoproteins have recently been found to play a critical role in the onset, progression, and prognosis of PD. ApoA-I demonstrates an important association with PD pathogenesis and consistent expression and reproducibility between blood fractions and various proteomic platforms, making it a valuable blood biomarker for PD. In addition, high levels of ApoA-I have shown a significant association with high-density lipoproteins, (HDL) correlating with a lower risk of PD and protecting against oxidative stress and neuroinflammation. However, repeated randomized control and observational studies are needed to investigate this further to provide more comprehensive findings that ApoA-I could be included as a potential blood biomarker of PD [58].

### 9.7. Potential Survey Biomarkers

A study by Lawton et al. [59] showed that it is possible to determine other potential blood biomarkers in the serum of patients with PD: vitamin D, uric acid, and C-reactive protein (CRP). These correlate with age and risk and severity of the disease; however, their effect should be subject to further examination.

## 10. Alzheimer’s Disease (AD)

AD is neuropathologically characterized by depositions of pathologically changed amyloid β-peptide in the form of amyloid plaques and hyperphosphorylated protein tau in the forms of neurofibrillary tangles. The pathogenic processes underlying AD affect synaptic functions in the initial, asymptomatic stage, long before cognitive decline and neurodegeneration. For this reason, reliable markers enabling early diagnosis and prognosis are necessary, especially for therapeutic intervention. Examining biomarkers in biological fluids has therefore mainly focused on the basic molecules of AD pathogenesis, namely amyloid beta (Aβ) and tau protein [7]. However, researchers are still looking for different analysis options from a range of other molecules and proteins that could help in clinical diagnosis.

Different approaches to the identification of blood biomarkers of AD can be classified as either those that focus on identifying specific molecules associated with known pathogenic mechanisms, or those that use omics methods to analyze the serum or plasma profile of molecules [60].

### 10.1. Amyloid Beta

Aβ levels, particularly Aβ_40_ and Aβ_42_, which are considered candidate biomarkers of AD, are being extensively studied and measured in patients’ plasma, but the correlation with their presence in the brain is null (of statistical but not clinical significance). Moreover, as mentioned in studies by Toombs and Zetterberg [61] and by Lopez et al. [62], plasma amyloid levels are influenced by several factors. The majority of amyloid in the peripheral circulation is primarily bound to albumin, acute phase proteins, and complement components. As a result, low albumin levels or multifactorial environmental conditions (e.g., inflammation) can lead to decreasing plasma availability of the albumin necessary to transport Aβ to the blood [63]. On the other hand, Janelidze et al. [64], found that to a certain extent, Aβ levels in the blood only reflect dysregulated Aβ metabolism degraded by circulating enzymes, or its metabolism in the liver which can potentially reduce monitoring of Aβ metabolism and its aggregation in the brain. Other factors unrelated to amyloid pathology may be modulated by peptide levels in the brain.

### 10.2. Hyperphosphorylated Tau Protein

Recent studies have reported that based on a blood test measuring one of the two forms of phosphorylated tau protein (p-tau181 or p-tau217), it is possible to differentially diagnose AD from another neurodegenerative disease with high accuracy [65,66]. These tests are able to identify the disease as early as in the stage of cognitive decline with tau and Aβ pathologies in the brain, with AUC (Area under the ROC curve) of up to 90%. The strong correlation between plasma p-tau181 and Aβ PET-positive and PET-negative individuals suggests that the new test detects AD pathology in the very early stages of the disease, even before signs occur, several years before the onset of dementia based on cognitive decline and hippocampal atrophy within one year [65]. However, the plasma p-tau181 level does not appear to increase any further in cases with moderate or severe tau pathology. From this perspective, it is a promising, available blood marker, specific for detecting AD pathology correlating with cerebral Aβ and tau pathology. Based on these findings, it can serve as a predictive marker of future cognitive decline, but its ability to monitor disease progression remains unclear [67,68]. In contrast, plasma p-tau217 is a confirmed dynamic biomarker of early stage AD, which may be useful for monitoring the disease progression in clinical practice and in treatment of AD [69].

### 10.3. cimiRNA and miRNA as Potential Biomarkers of AD

Among other research results, a group of non-coding RNAs, “circulating microRNAs” (cimiRNA), seem to be a promising biomarker. This is a newly identified class of single-stranded RNAs formed by head-to-tail splicing (back splicing). cimiRNA is produced in the nucleus and consequently transported into cytoplasm, from which it is either actively or passively transported into bodily fluids. cimiRNAs are, due to a lack of loose ends and a half-life longer than 48 h, extremely stable and resistant to degradation by endogenous RNA activity [70]. They are also able to withstand demanding environmental conditions, such as extreme pH values, boiling, long-term storage, and multiple defrost cycles. The advantage is that they can be detected by quantitative methods, such as Real-Time Quantitative Reverse Transcription PCR (qRT-PCR), and by semiquantitative methods, such as next-generation sequencing (NGS). Until recently, it was assumed that cimiRNAs that correlate with AD-associated genes such as *APP* (amyloid precursor protein), *PSEN1* (presenilin 1), and *PSEN2* (presenilin 2) play a key role in AD pathophysiology. However, some cimiRNA groups, specifically miR-9-5p, miR-125-5p, miR-146a-5p, and miR-155-5p have been proven to be upregulated in patients before the observed correlation between Aβ levels and AD pathophysiology significantly increased. This emphasizes the potential usefulness of cimiRNAs as biomarkers for an early diagnosis of AD [71].

It appears that miRNAs regulate synaptic homeostasis and plasticity processes. This indicates their ability to be involved in early synaptic dysfunction during AD. Therefore, miRNAs have been described as promising, cost-effective, and non-invasive biomarkers of AD, capable of detecting asymptomatic stages of the disease [72].

### 10.4. Potential Blood Biomarkers

Blood neutrophils, brain-derived neutrophil factors, and TNF receptor superfamily member 13C (*TNFRSF13C*) and chemokine stroma-derived factor 1 (*CXCL12*) genes have been identified as additional potential blood biomarkers of AD as their expression levels were independent risk factors. Expression of these β-cell-related genes was increased in peripheral blood and correlated with patients’ tau and Aβ in CSF.

*TNFRSF13C* codes B cell-activating factor receptor (BAFFR) during ontogenesis, terminal differentiation, maturation, and survival of β-cells. Wu et al. [73] showed that *TNFRSF13C* downregulation and overexpression of *CXCL12* contribute to the confirmation of AD diagnosis; the number of β-cells can also play an important role in the etiology of the disease.

### 10.5. Flotillin as a Diagnostic Marker

Abdullah et al. conducted a research project investigating abundant exosomal protein flotillin, which may serve as a new diagnostic marker of AD [74]. Serum flotillin levels were found to be significantly decreased in patients with AD, as compared to controls. The decreased concentration levels were explained by the authors as reduced exosomal secretion caused by Aβ_42_ oligomeric forms. It appears that it could be a molecule with a secondary response to Aβ_42_ pathogenesis [7].

### 10.6. Sphingolipids and AD

The presence of four sphingolipids (sphingomyelin (SM) with acyl residue sum SM C16:0, SM C18:1, SM C16:1 and hydroxysphingomyelin with acyl residue sum SM C14:1) that Varma et al. [75] described in their study, proved that their higher concentrations in blood in cognitively normal individuals are associated with a higher risk of future conversion to incident AD, in the prodromal and preclinical stages of the disease. Changes in blood levels of these phospholipids and sphingolipids may reflect a metabolic disorder and/or neuronal degeneration in CNS in the early stages of the disease without advanced cognitive signs. However, it is necessary to continue researching whether they are able to distinguish AD from other types of dementia.

### 10.7. Neurofilament Light Chains

The last suitable object of current research in patients’ plasma are NFLs, which could represent a non-invasive biomarker of neurodegeneration in AD. However, to analyze them from serum, it is necessary to use very sensitive methods such as SIMOA or electrochemiluminescence [76].

Their action is associated with hypometabolism and atrophy. Their levels rise in concentration together with hyperintensity of cerebral white matter. NFLs have also been found to correlate with patient age and total tau protein levels (t-tau) and hyperphosphorylated tau protein (p-tau), yet it does not correlate with *ApoEε4* genotype or patient sex. However, it may influence the prediction of cognitive impairment, especially in the preclinical stages of AD [77]. These findings indicate that plasma NFL is a general marker of neuronal degeneration but not specific to AD. Therefore, we can argue that NFL is a dynamic biomarker that can be used to measure the intensity and progression of ongoing neurodegeneration while it changes in the course of AD [78,79].

## 11. Conclusions

The importance of biomarkers in diagnosing neurodegenerative diseases is a very attractive and pursued topic. Determining a reliable biomarker that could assist in the unambiguous identification of clinical-diagnostic units or differentiating disease manifestations with more proteinopathies, including their course and therapy, is the first step to successful treatment of these diseases.

Currently, diagnostical practice relies on a range of identified, routinely biochemically examined biomarkers or focused on the discovery of blood biomarkers bearing the major characteristics of neurodegenerative diseases (Table 2), which correlate, to a large extent, to levels of CSF biomarkers.

The high heterogeneity of neurodegenerative diseases requires more accurate and clearer biomarkers that are able to help improve clinical therapy. For this reason, a number of studies have focused on finding them. The process of protein misfolding and aggregation appears years or decades before the onset of clinical symptoms.

However, there are still limitations in the use of less-sensitive methods, and obsolete diagnostic criteria that should be replaced or combined with ultrasensitive methods such as SIMOA or electrochemiluminescence, which are able to analyze serum protein aggregates with high sensitivity [15,20]. The ability to quantify every aspect of disease at high-definition and through blood tests that could identify a disease, even before the first signs occur, is a particularly important aspect of research, especially because of the need for early intervention for successful treatment. Additionally, analyses of exosomes as active carriers, molecules involved in intraneuronal and neuronal glial communications, or metabolite analyses should be considered in the future.

CSF biomarkers still represent the gold standard for the basic molecular characterization of disease and the identification of biologically homogeneous groups of patients. Unfortunately, blood biomarkers are not yet able to completely replace CSF biomarkers, although some tests have had very promising results and, moreover, were approved by the U.S. Food and Drug Administration. Yet in the future, blood tests may become the first screening for defining target engagement and a tool for the long-term observation of treatment results, followed by CSF analysis.

A consensus has not yet been reached in the literature; there are still many gaps and discrepancies. Therefore, a clearer understanding of which diagnostic stage each biomarker is useful in, of their interrelations with other examination methods, and of the relationships between diseases is necessary. Additionally, it would be appropriate to implement specific biomarker patterns into the diagnostic criteria of individual proteinopathies. Additionally, we should consider the incongruence of the clinical and pathological picture, where the markers should be particularly useful [80]. Moreover, it would be useful to introduce examination biomarkers from serum, which is a more available biological fluid than CSF, and is a much less invasive intervention for patients.

## Figures and Tables

**Figure 1 biomedicines-10-01760-f001:**
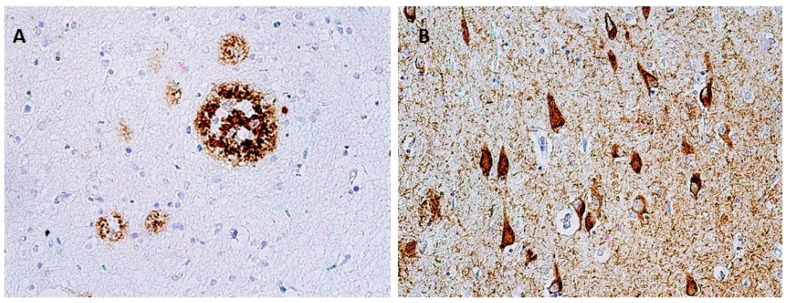
Neuropathological finding showing changes in Alzheimer’s disease. (**A**) Senile plaques in temporal cortex, immunohistochemically stained with β-amyloid antibodies, magnification 400×; (**B**) Neurofibrillary tangles, pretangles, and numerous threads in hippocampus, stained with monoclonal antibody (AT8) against phosphorylated tau protein, magnification 400×.

**Figure 2 biomedicines-10-01760-f002:**
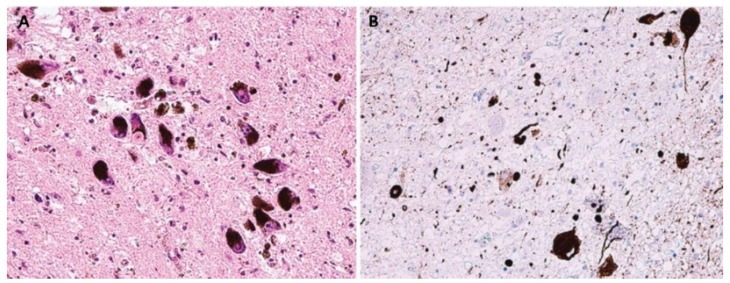
Neuropathological finding showing changes in Parkinson’s disease. (**A**) Lewy body in a pigmented neuron substantia nigra, H&E staining, original magnification 200×; (**B**) The spectrum of α-synuclein pathology in pons. Lewy bodies, neuronal granular cytoplasmic positivity and dystrophic neurites, stained with anti- α-syn monoclonal antibody, original magnification 200×.

**Figure 3 biomedicines-10-01760-f003:**
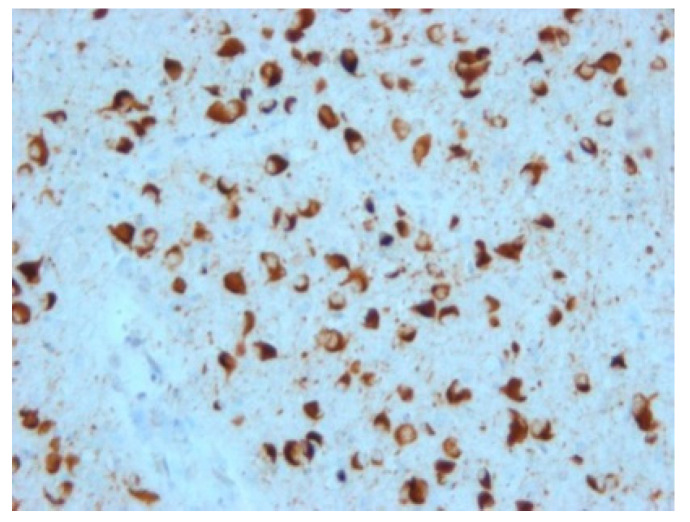
Neuropathological finding showing changes in multiple system atrophy. Typical alpha-synucleinopathy with oligodendroglial cytoplasmic inclusions in striatum, stained with anti- α-syn monoclonal antibody, original magnification 200×.

**Figure 4 biomedicines-10-01760-f004:**
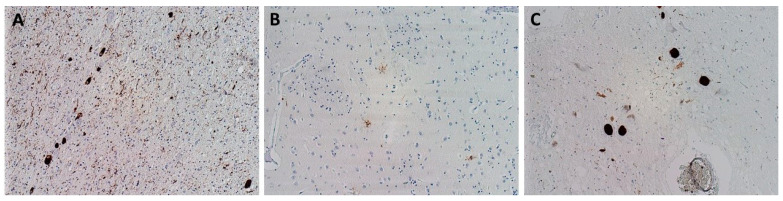
Tau pathology spectrum in various brain areas in progressive supranuclear palsy. (**A**) Numerous threads, coiled bodies, and neurofibrillary tangles in subthalamic region, stained with monoclonal antibody (AT8) against hyperphosphorylated tau protein; (**B**) Tufted astrocytes in the basal ganglia, stained with monoclonal antibody (AT8) against pathologically changed tau protein; (**C**) Neurofibrillary tangles in the substantia nigra neurons, stained with monoclonal antibody (4RD) against tau protein, original magnification 100×.

**Figure 5 biomedicines-10-01760-f005:**
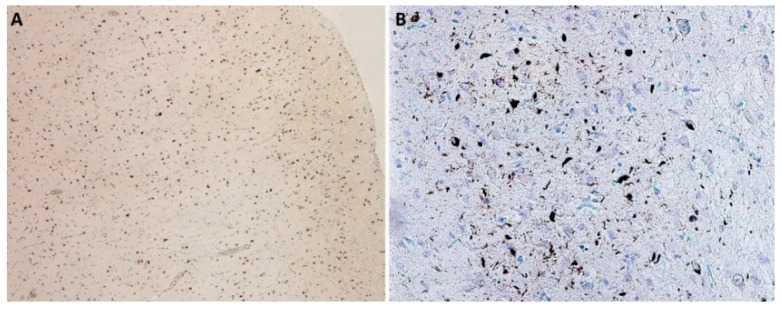
Neuropathological finding showing changes in FTLD-TDP. (**A**) Temporal cortex, numerous neuronal cytoplasmic inclusions in all layers of neocortex, stained with monoclonal antibody (phospho-TDP-43) against hyperphosphorylated TDP-43 protein, original magnification 40×; (**B**) cytoplasmic inclusions in residual neurons *nuclei nervi hypoglossi*, stained with monoclonal antibody (phospho-TDP-43) against pathologically changed TDP-43 protein, magnification 200×.

**Figure 6 biomedicines-10-01760-f006:**
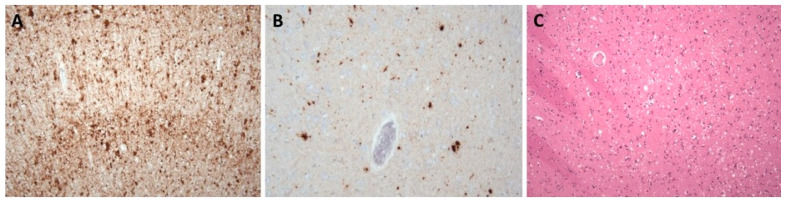
Neuropathological finding showing changes corresponding to Creutzfeldt–Jakob disease. (**A**) Diffuse synaptic positivity, enhanced perivacuolar “patchy” positivity and “plaque-like” positive structures in cerebral cortex, stained with monoclonal antibody (12F10) against prion protein, original magnification 200×; (**B**,**C**) diffuse synaptic positivity and enhanced perivacuolar “patchy” positivity and “plaque-like” positive structures in striatum, stained with monoclonal antibody (6H4) against prion protein, original magnification 200× and H&E staining, original magnification 100×.

**Table 1 biomedicines-10-01760-t001:** CSF biomarkers for individual groups of neurodegenerative diseases.

CSF Biomarker		α-Synucleinopathies				Tauopathies					Prion Diseases
		Lewy Body Diseases				“Primary Tauopathies” FTLD-Tau				Non-Tau FTLD	
	AD	PD	PDD	DLB	MSA	FTD	Pick’s Disease	PSP	CBD	TDP-43 Proteinopathies	
α-synuclein (t-α-syn)		+	+	+	+						
o-α-syn		+		+							
p-α-syn		+									
o-α-syn/t-α-syn		+									
tau protein (t-tau)	+	+	+	+	+	+	+	+	+	+	+
p-tau	+	+	+	+	+	+	+		+	+	+
t-tau/p-tau						+					+
t-tau/α-synuclein				+							
Aβ_42_	+	+	+	+		+	+	+		+	
Aβ_40_	+			+							
Aβ_42_/Aβ_40_	+		+	+							
t-tau/Aβ_42_	+					+					
p-tau/Aβ_42_	+			+		+					
NFL	+	+			+	+		+	+		+
NFH									+		
14-3-3 protein											+
t-tau/14-3-3											+
TDP-43						+				+	+
pTDP-43						+				+	
TDP-43/Aβ_42_	+									+	
dopamine/DOPAC		+				+					
UCH-L1		+			+			+	+		
GBA		+									
YKL-40	+	+									
VILIP-1	+										
TREM2	+										
DJ-1	+	+			+						
SNAP25	+										
BACE1	+										
neurogranin	+										
chromatogranin A		+			+						
cystatin C	+										
clusterin	+	+	+	+	+						
progranulin						+				+	
DPRs						+				+	
GFAP	+	+									
MCP-1	+										
miRNA		+									
HVA		+									

This table summarizes elementary CSF biomarkers for main groups of neurodegenerative diseases; “+” symbol denotes usable markers for individual ND that directly characterize a disease or assist in their mutual distinction. Other shown biomarkers without a symbol do not have any significant clinical use in diagnostics of the given disease. t-α-syn—total α-synuclein; o-α-syn—oligomeric form of α-synuclein; p-α-syn—phosphorylated form of α-synuclein; t-tau—total tau protein; p-tau—phosphorylated tau protein; Aß42—amyloid beta, hyperphosphorylated isoform 42; Aß40—amyloid beta, hyperphosphorylated isoform 40; NFL—neurofilament light chains; NFH—neurofilament heavy chains; TDP-43—TAR DNA-binding protein 43; pTDP43—phosphorylated TAR DNA-binding protein 43; DOPAC—dihydroxyphenyl acetate; UCH-L1—ubiquitin C-Terminal Hydrolase-L1; GBA—β-glucocerebrosidase; YKL-40—chitinase-3-like protein 1; VILIP-1—visinin-like protein 1; TREM2—myeloid cells 2; DJ-1—Daisuke-Junko-1; SNAP25—synaptosomal associated protein; BACE1—β-site APP-cleaving enzyme 1; DPRs—Dipeptide repeat proteins; GFAP—Glial fibrillary acidic protein; MCP-1—Monocyte chemoattractant protein-1; miRNA—microRNA; HVA—homovanillic acid.

**Table 2 biomedicines-10-01760-t002:** Overview of blood biomarkers in PD and AD.

** *Parkinson’s Disease* **	** *Inclusion of Blood-based Biomarkers* **
α-synuclein	predictive biomarker
urate	diagnostic-prognostic biomarker
clusterin	predictive biomarker
miRNA-124	potentially diagnostic biomarker
NFL	predictive biomarker
ApoA-I	potentially predictive biomarker
vit. D, CRP, uric acid, ITIH4, transthyretin	potential biomarker
** *Alzheimer’s Disease* **	** *Inclusion of Blood-based Biomarkers* **
Aβ	predictive biomarker
ptau-181	diagnostic-predictive biomarker
ptau-217	diagnostic-prognostic biomarker
cimiRNA	diagnostic biomarker
sphingolipids	prognostic biomarker
NFL	dynamic biomarker
flotillin	diagnostic biomarker
neutrophils, neutrophil factors, genes *TNFRSF13C*, *CXCL12*	potential biomarkers

This table summarizes an overview of biomarkers usable in serum diagnosis in Parkinson’s and Alzheimer’s diseases. The individual biomarkers are divided in relation to the type and applicability of the biomarker in clinical diagnosis. miRNA-124—microRNA-124; NFL—neurofilament light chains; ApoA-I—apolipoprotein A-I; CRP—C reactive protein; ITIH4—Inter-α-trypsin inhibitor heavy 4; Aβ—amyloid beta; ptau-181—phosphorylated protein at position 181; ptau-217—phosphorylated protein at position 217; cimiRNA—circulating micro RNA; TNFRSF13C—TNF receptor superfamily member 13C, CXCL12—chemokine stroma-derived factor 1.

## Data Availability

Not applicable.

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
