# Peer review of "Biomarkers of Neurodegenerative Diseases: Biology, Taxonomy, Clinical Relevance, and Current Research Status"

_biomedicines, 2022, doi:10.3390/biomedicines10071760_

Round 1

Reviewer 1 Report

I think the work done by the authors is relevant, but considering the title "Biomarkers of neurodegenerative diseases: Biology, taxonomy, clinical relevance, and current research status," I invite the authors to improve this Review by arguing other neurodegenerative diseases such as Gerstmann-Sträussler-Scheinker Disease (GSS) and Amyotrophic Lateral Sclerosis (ALS).

Author Response

Point 1: I think the work done by the authors is relevant, but considering the title "Biomarkers of neurodegenerative diseases: Biology, taxonomy, clinical relevance, and current research status," I invite the authors to improve this Review by arguing other neurodegenerative diseases such as Gerstmann-Sträussler-Scheinker Disease (GSS) and Amyotrophic Lateral Sclerosis (ALS).

Response 1: This review was improved by arguing other neurodegenerative diseases such as Gerstmann-Sträussler-Scheinker Disease (GSS) (P5, sections 2,3) and Amyotrophic Lateral Sclerosis (ALS) (P4, section 7).

Reviewer 2 Report

The manuscript entitled “Biomarkers of neurodegenerative diseases: Biology, taxonomy, clinical relevance, and current research status” is a fine paper with good structure that summarizes biomarkers of neurodegenerative diseases, especially cerebrospinal fluid biomarkers and biomarkers in blood serum. There are, however, some areas in which the paper can be improved.

Introduction on neurodegenerative diseases.
Table 1, Align the neurodegenerative diseases with the main text.
Combine some short paragraphs in the main text. For example, P8, section 4.

Author Response

Point: The manuscript entitled “Biomarkers of neurodegenerative diseases: Biology, taxonomy, clinical relevance, and current research status” is a fine paper with good structure that summarizes biomarkers of neurodegenerative diseases, especially cerebrospinal fluid biomarkers and biomarkers in blood serum. There are, however, some areas in which the paper can be improved.

1: Introduction on neurodegenerative diseases.
2: Table 1, Align the neurodegenerative diseases with the main text.
3: Combine some short paragraphs in the main text. For example, P8, section 4.

Response:

1: It was improved introduction on neurodegenerative diseases.

2: Table 1 was modified, align the neurodegenerative diseases with the main text.

3: It was combined some short paragraphs in the main text (P1, section 2; P4, section 2; P7, section 6; P8, section 4; P9.4, section 2; P 9.6, section 3; P 10.7, section 3)

This manuscript is a resubmission of an earlier submission. The following is a list of the peer review reports and author responses from that submission.